# ENTROPY PROXY FOR LLM MEMORIZATION SCORE

## ABSTRACT

Large Language Models (LLMs) are known to memorize portions of their training data, sometimes reproducing content verbatim when prompted appropriately. Existing memorization research rarely explores how training data influences memorization and often limits the experimental setup to a binarized memorization vs non-memorization catagory. In this work, we investigate a fundamental yet under-explored question in the domain of memorization: *How to quantitatively characterize memorization difficulty using intrinsic properties of training data in LLMs?* Inspired by early studies using compression algorithms to filter out simple memorization cases, we explore the link between training data compressibility and memorization. Through experiments on a wide range of open models without various setups, we present the Entropy–Memorization Law. It suggests that at the set-level, data entropy (estimator) is linearly correlated with memorization score. We also further investigate EM Law with several dimensions: visualizing vocabulary size as an implicit factor, and applying the law to data with disparate semantics.

## 1 INTRODUCTION

Large Language Models (LLMs) demonstrate remarkable performance in capturing linguistic patterns and generating coherent text (Vaswani et al., 2017; Radford et al., 2019). It is sweeping across every domain in natural language processing. Alongside dominating performance through various benchmarks, a critical phenomenon has emerged: LLMs are shown to memorize and reproduce verbatim sequences from their training corpora (Carlini et al., 2019; 2020). This memorization behavior has raised growing concerns, particularly in terms of privacy leakage and intellectual property protection. For example, studies have shown that LLMs can inadvertently generate personally identifiable information (PII) (Carlini et al., 2020), or proprietary data from books (USAuthorsGuild, 2023; LLMLitigation, 2023) and news articles (Michael, 2023). Most recently, Anthropic reached a USD 1.5 billion settlement with authors over the unauthorized use of copyrighted books, underscoring the growing legal risks surrounding LLM training data (The New York Times, 2025).

As scaling laws (Kaplan et al., 2020) drive LLM developers to expand model capacity and training data for performance improvements, research (Wang et al., 2025; Ippolito et al., 2023) has demonstrated that memorization scales with model size. Broader data exposure in LLM training elevates the risk of leakage for all internet-sourced content. Furthermore, memorization is shown to be necessary for generalization (Feldman & Zhang, 2020) Therefore, advancing the theoretical understanding of the factors that shape memorization has become a crucial and urgent issue in LLM development. Factors can be categorized into three types: model training paradigms, test-time compute (i.e. prompting strategies), and training data. However, existing literature is limited in two aspects:

First, previous memorization studies mainly focus on prompting strategy (Carlini et al., 2020; Schwarzschild et al., 2024), and training paradigm (Chu et al., 2025). The role of training data in memorization is under-explored. Regarding this topic, the existing research limits the scope to duplicated data, where researchers find out that data duplication significantly increases memorization (Kandpal et al., 2022; Biderman et al., 2023b). There is a lack of systematic investigation on how intrinsic properties of training data affect memorization. We argue that as the core of data-driven Machine Learning methods, training data should be explored in depth.

Second, most existing memorization explorations are limited to *qualitative* studies. Most research work typically regards memorization within a binary framework Carlini et al. (2020); Zhang et al.

(2023); Prashanth et al. (2025); Schwarzschild et al. (2024). The only well-known quantitative study is by Carlini et al. (2023), where researchers reveal three log-linear relationships on three properties: model scale, data duplication, and context length. However, these properties are not intrinsic to training data.

Concerning the above limitations, this paper addresses an open and under-explored question: ***How to characterize memorization difficulty of training data in LLMs quantitatively?***. This paper formulates memorization difficulty using an integer-valued memorization score. With a proper definition of memorization score, this paper then seeks metrics directly related to training data that approximate the memorization score. In this way, we quantitatively explore the relationship between memorization score and training data.

Specifically, this work explores the possibility of adopting **compressibility** of training data as our proxy. Such a choice is inspired by early memorization work Carlini et al. (2020), where researchers use text compressors to filter out "simple" memorization cases, e.g., counting from 1 to 100. From another perspective, the community may implicitly assume that highly compressible text corresponds to lower memorization difficulty, but it is not examined formally and rigorously. On the other hand, entropy is closely related to compressibility. The source coding theorem (Shannon, 2001) states that the entropy of a distribution provides the fundamental lower bound on the average code length achievable for samples drawn from that distribution.

In this paper, inspired by the above facts, we apply two metrics related to compressibility: zlib compression (Deutsch & Gailly, 1996; Carlini et al., 2020) and entropy (estimators). We consider metrics adopted at two levels: instance-level and set-level. We empirically show that the set-level entropy estimator approximates the memorization score well. Measuring fitness by linear regression, we achieve $r > 0.9$ across a wide range of popular LLMs. We dub this core finding of the study as **Entropy–Memorization Law**. It suggests that higher entropy correlates with a higher memorization score in LLMs.

Our entropy estimator enjoys twofold benefits: 1) the metric gives a quantitative description of memorization. The quantitative metric advances beyond the traditional binary setting of memorization with qualitative empirical observations. A quantitative metric facilitates the assessment of privacy risks for LLM providers. 2) the metric is model-agnostic. A model-agnostic is compute-efficient. It does not require back propagation with a large number of model weight. In contrast, model-aware approaches, like influence functions (Koh & Liang, 2017; Feldman & Zhang, 2020) typically require hessian computation or even re-training, which is not affordable on LLMs.

EM Law is empirically validated on a wide range of pre-trained models, including the OLMo family (Groeneveld et al., 2024), OpenLlama (Geng & Liu, 2023), and Pythia (Biderman et al., 2023b). We also explore EM Law in various experimental setups, including continuation length, inference sampling strategy.

We conduct thorough investigations into EM Law under several dimensions. **First**, we consider an implicit factor that shapes EM Law: the support set over which entropy is defined. We identify that lower memorization-score data comprises *exponentially-linear* fewer unique tokens, and achieves *linearly* higher entropy values given the support size. **Second**, we discuss EM Law under data with different semantics, and it turns out that the slope and intercept of the resulting regressed line exhibit a significant difference under different semantic data.

## 2 EXPERIMENTAL SETUP

**Threat Model** This paper assumes a hypothetical engineer who studies the characterization of training data on an LLM. Therefore, it is necessary for the engineer to have full access to the LLM *and* its training data. This engineer freezes other potential confounders to the memorization score, including prompt strategy, and training paradigm.

This paper studies *pre-trained*-only LLM, (i.e. "base" version of LLMs), since post-training may involve different model training paradigms, bringing uncontrollable noises to our experiments.

**Choices of LLM and Training Corpus** We select three state-of-the-art open-data LLMs: OpenLlama (Geng & Liu, 2023),

We selected four family of pre-trained only LLMs: (1) OLMo (Groeneveld et al., 2024) pre-trained on Dolma (Soldaini et al., 2024) dataset, and (2) OLMo-2 (OLMo et al., 2024) pre-trained on OLMo-2-1124-Mix (OLMo et al., 2024) dataset; (3) OpenLlama (Geng & Liu, 2023) pre-trained on Redpajama (Computer, 2023); (4) Pythia (Biderman et al., 2023b) pre-trained on the Pile (Gao et al., 2020).

**Prompting Strategies**    In this study, we consider a *Discoverable Memorization* (DM) scenario (Nasr et al., 2025; Carlini et al., 2023; Kandpal et al., 2022; Ippolito et al., 2023). Formally, DM denotes the following: we sample $N$ token sequences from the training dataset. Each sequence is partitioned into $(p, s)$, where $p$ (first $|p|$ tokens) serves as the prompt and $s$ (remaining tokens) serves as the answer for model $\theta$. Afterwards, LLM $\theta$ generates response $r = \theta(p)$. The memorization score measures the difference between two sequences $r$ and $s$. By default, we set $|p| = 100$ and $|s| = |r| = 50$, following the popular setup in the community (Al-Kaswan et al., 2023). However variants of continuation length will be discussed in Section 5. We defer the detailed statistics of the sampled datasets to Appendix A.

The token sequence sampling strategy is as follows: we repeatedly randomly sample a sequence (length > $|p + s|$) from the dataset until the number reaches the required number.

**Filtering Trivial Memorization**    We exclude *trivial* memorization cases that fall outside the scope of our interest: the LLM response $r$ may exhibit highly lexical overlap with the prompt $p$. For instance, LLM may copy a long URL from the prompt as the response. Although such a response may match the answer, however, such a match should be attributed to the prompt, instead of the memorization capacity of LLM. To conduct filtering, we design a Longest Common Subsequence (LCS)-based filtering method. We establish a thresholding strategy based on LCS: samples for which $LCS(p, s) \geq \frac{|s|}{2}$ are excluded from further memorization analysis, while samples below the threshold are retained.

**Memorization Score**    A *binary* classification does not provide quantitative description on memorization. We therefore need a memorization score $d(r, s)$, measuring the differences between response $r$ and answer $s$ at the token level. Following previous memorization work (Dong et al., 2024), we uses edit distance (Levenshtein et al., 1966), defined by the minimal number of single-token edit operations – insertions, deletions, and substitutions – required to transform one sequence into another. A *higher* memorization score indicates *lower* similarity between two sequences.

Depending on the notions of memorization, other memorization scores are adopted in the community. The traditional one is *exact match* (Carlini et al., 2023; Tirumala et al., 2022; Carlini et al., 2019; 2020), is a binary decision, which is limited. Another line of research uses semantic similarity, measuring the similarities based on sentence embeddings generated by LMs (Reimers & Gurevych, 2019; Chen et al., 2024). However, memorization at the semantic level could be subjective. Therefore, Levenshtein distance is a good fit as a memorization score in our study.

To summarize, the research question in this paper is formulated as follows:

> **Assumption.** A fixed pre-trained LLM $\theta$, a fixed prompting strategy $DM$ to generate $p$, and a memorization score $d(r, s) = d(\theta(p), s)$.
> **Goal of the study.** Find an approximator function $M(s)$ of memorization score $d(r, s)$.

## 3    THE FIRST ATTEMPT: INSTANCE-LEVEL COMPRESSIBILITY

How to characterize training data $s$? Inspired by the history of using compression algorithms to filter out simple memorization cases, we explore the link between training data compressbility and memorization. We suspect that higher compression rate answers exhibit greater randomness, and may be harder to be memorized.

Regarding compressibility, we employ two metrics: zlib compression ratio, and an estimated entropy [1] (Deutsch & Gailly, 1996) based on empirical point probabilities (Carlton, 1969).

---

[1]We assume a base-2 logarithm for all entropy calculations throughout the work.

Two metrics correspond to two types of lossless coding algorithms in information theory: coding schemes for sources with memory and without memory, i.e., memoryless.

Both metrics are *instance-wise*, i.e., calculated per sequence. Here are the formal notations to describe the calculation. An instance $s_i = (s_i^1, s_i^2, ..., s_i^{|s_i|})$ is a sequence, where $s_i^j$ is a token. All tokens within $s_i$ form the sample space $\mathcal{T}_i$. Then for each token $x \in \mathcal{T}_i$, the empirical point probabilities $\hat{p}_i(x)$ are calculated as:

$$\hat{p}_i(x) = \frac{1}{|s|} \left| \{j \mid s_i^j = x\} \right|. \tag{1}$$

$\hat{p}_i(x)$ is the relative frequency of $x$ in the observed sequence. In this attempt, we use entropy estimated by the empirical point probabilities as our approximator $M(s_i)$:

$$M(s_i) \triangleq - \sum_{x \in \mathcal{T}_i} \hat{p}_i(x) \log \hat{p}_i(x) \tag{2}$$

In practice, the distribution for language is unknown. we instead learn from samples. The above estimator approximates entropy by viewing the point probability as samples from the empirical distribution itself.

With the established $M(s_i)$, we are interested in whether $M(s_i)$ is a good approximator of the memorization score $d(r_i, s_i)$. To achieve this, we gather all $(M(s_i), d(r_i, s_i))$ pairs obtained by empirical observations in a scatter plot, and further study their correlation. The detailed algorithm is as follows:

---

**Algorithm 1:** Instance-wise entropy estimator.

**Input:** LLM $\theta$, and its training corpus $D$
**Output:** Plot of $(d(r_i, s_i), M(s_i))$

1  Sample $N$ prompt-answer pairs $\{(p_i, s_i)\}$ from $D$;
2  **for** $i \leftarrow 0$ **to** $N-1$ **do**
3      $r_i \leftarrow \theta(p_i)$;
4      $\hat{p}_i \leftarrow \text{EmpProb}(r, s)$ // Eq. 1
5      $M(s_i) \leftarrow$
        $- \sum_{x \in \mathcal{T}_i} \hat{p}_i(x) \log \hat{p}_i(x)$ // Eq. 2
6      $d(r_i, s_i) \leftarrow d_{\text{lev}}(r_i, s_i)$
7      Plot $(d(r_i, s_i), M(s_i))$;
8  **end**

---

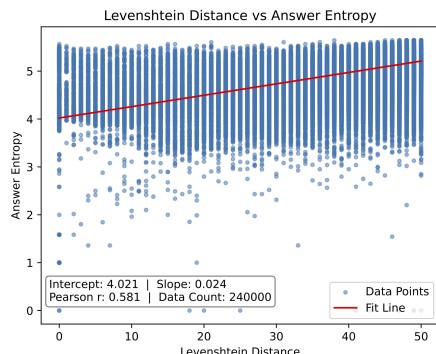

Figure 1: Memorization score *v.s.* entropy estimator observed on OLMo-1B.

We run algorithm 1 on the OLMo-1B model, and obtain the scatter plot as illustrated in Fig. 1. A linear regression based on ordinary least squares is conducted on the plot.

The regression analysis suggests a positive linear relationship between Levenshtein Distance-based memorization score and our approximator. However, the observed pattern is very noisy, as demonstrated by the weak Pearson correlation $r = 0.581$. Zlib is also shown to fit poorly, where we defer the details to Appendix B.1.

We regard the failure of our first attempt as due to the limitation of sample space. In our experimental setup, an LLM generates up to $|s_i| = 50$ tokens. Hence, the sample space size $|\mathcal{T}_i|$ is upper bounded by 50 in ideal cases. This space is orders of magnitude *smaller* than the full token space. In reality, the full token space is defined by the tokenizer vocabulary, and the vocabulary size of the OLMo-1B tokenizer is about 50k. Therefore, our entropy estimation is too noisy to reflect the real-world scenarios. Our second attempt addresses the limitation and will be discussed in the next section.

> **Summary.** Both zlib and the entropy estimate fail to fit memorization score at instance-level, since sample space is orders of magnitude smaller than the full token space.

## 4   THE SECOND ATTEMPT: SET-LEVEL COMPRESSIBILITY

For the entropy estimator, we regard that our first attempt's failure is attributed to insufficient sample space for entropy estimation. The second attempt addresses this by substantially expanding the sample space. We observe that in the first attempt (Fig. 1), for a fixed memorization score, we constructed more than one (even thousands of) estimates. Then an intuitive idea is to aggregate these estimates to a new robust estimate. Specifically, we expand the sample space from tokens with *one* instance, to *all* the instances with the same memorization score. The idea is the same as building "level-sets" of memorization score. Mathematically, for a fixed memorization score $e$, the new sample space

$$\mathcal{T}_e = \bigcup \left\{ s_i^j \mid d(r_i, s_i) = e, i \in \{0, \cdots, N-1\}, j \in \{0, \cdots, |s|-1\} \right\}. \tag{3}$$

Then the empirical probabilities $\hat{p}_e(x)$ are calculated within new sample space $\mathcal{T}_e$:

$$\hat{p}_e(x) = \frac{1}{N|s|} \left| \{(i, j) \mid s_i^j = x, d(r_i, s_i) = e\} \right|. \tag{4}$$

**Algorithm 2:** Level-set-wise entropy estimator.

**Input:** LLM $\theta$, and its training corpus $D$
**Output:** Plot of $(e, M(s_e))$
1  Sample $N$ prompt-answer pairs$\{(p_i, s_i)\}$ from $D$;
2  **for** $i \leftarrow 0$ **to** $N-1$ **do**
3  |   $r_i \leftarrow \theta(p_i)$;
4  |   $d(r_i, s_i) \leftarrow d_{\text{lev}}(r_i, s_i)$
5  **end**
6  **for** $e \leftarrow 0$ **to** $|s|-1$ **do**
7  |   $\hat{p}_e \leftarrow \text{NewEmpProb}(r, s)$ // Eq. 4
8  |   $M(s_e) \leftarrow -\sum_{x \in \mathcal{T}_e} \hat{p}_e(x) \log \hat{p}_e(x)$;
9  |   Plot $(e, M(s_e))$.
10 **end**

The new empirical probabilities benefit from a larger sample space. Similar to Eq. 2, we then use new empirical probabilities to derive a new level-set-based entropy estimate to approximate the memorization score $e$.

$$M(s_e) \triangleq - \sum_{x \in \mathcal{T}_e} \hat{p}_e(x) \log \hat{p}_e(x) \tag{5}$$

The modified algorithm is as shown in Alg. 2. It turns out that the level-set-based entropy estimator is shown to be a good approximator of memorization score, as demonstrated in Figure 2 below.

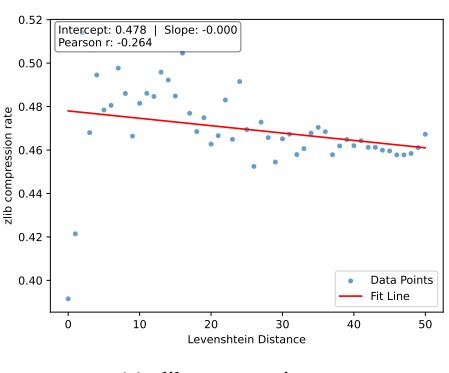

(a) zlib compression rate.

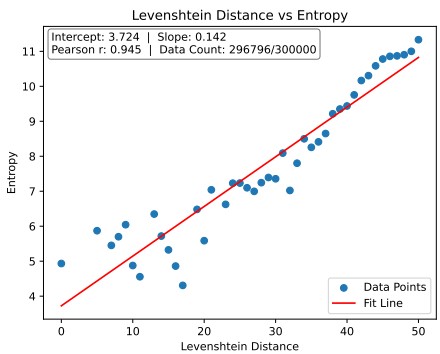

(b) Level-set-based Entropy Estimator.

Figure 2: Comparison of set-level compressibility metrics on OLMo-2-1124-7B.

At set-level, entropy estimator outperforms zlib compression rate on memorization score approximation.

## 5   ENTROPY–MEMORIZATION LAW

We run algorithm 2 on an extensive range of open-dataset LLMs, present the empirical results in Fig. 3. We observe very strong linear empirical results ($r = 0.972$ and $0.945$ respectively) on both

plots. It indicates that **the level-set-based entropy estimator is an effective *linear* approximation of memorization score**. We name this discovery as *Entropy-Memorization Law*. Additional results are available in Appendix B.3.

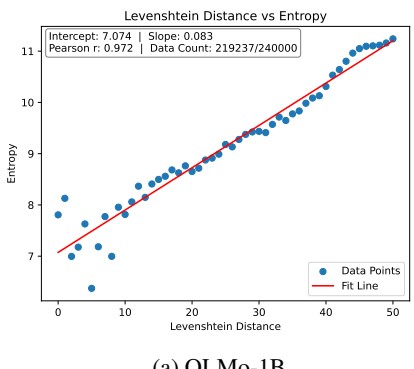

(a) OLMo-1B

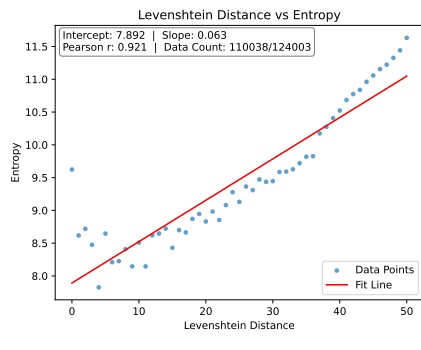

(b) Pythia-70m-deduped

Figure 3: Entropy–Memorization Law on open-dataset LLMs.

**Entropy–Memorization Law is preseved under varying continuation lengths** We explores different continuation token lengths, including $\{10, 20, 30, 40, 50\}$. As a demonstration, we use OLMo-2-1124-7B and its training dataset OLMo-2-1124-Mix in this experiment. Note that we rescaled the memorization score to the range $[0, 1]$ in the plot for better presentations.

Figures 4 present the experimental result when generation token length varies. The Pearson correlation coefficients ($r$) remain very high (ranging from 0.92 to 0.98) across all settings, indicating robust fitness under varying token lengths. Although regression lines have different slopes and intercepts, the $y$-value reaches about 11 when the memorization score $e = 50$. Besides, $M(s_{50}) \approx 11$.

Another observation is that there is a monotonic increase in the intercept values of the fitted regression lines as the generation token length increases. This matches our expectations derived from information theory. Denote the

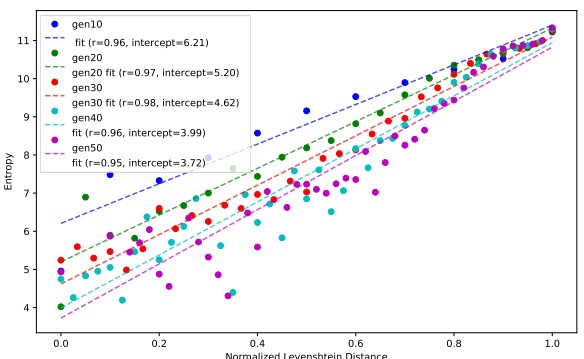

Figure 4: Entropy–Memorization Law under varying generation token lengths.

vocabulary size as $|V|$, $n$-sequence has $|V|^n$ potential outcomes; hence when $n$ gets larger, the maximum entropy of $n$-sequence increases.

In appendix B.2, we also explore various LLM inference sampling strategy, including temperature, top-p and and top-k sampling.

### 5.1 NORMALIZED ENTROPY–MEMORIZATION LAW

In our framework, entropy is determined by two key factors: (1) the cardinality of the possible outcomes (sample space size), and (2) the distribution of empirical probabilities across these outcomes. Mathematically, given $n$ possible outcomes, the entropy of a discrete random variable (r.v.) is upper bounded by $-\sum_{i=1}^{n} \frac{1}{n} \log \frac{1}{n} = \log n$. The upper bound is achieved when *r.v.* is uniformly distributed. Hence, a two-outcome *r.v.* has maximum entropy $\log 2 = 1$ bit, while a *three*-outcome *r.v.* yields maximum $\log 3$ bit entropy. Then, we are interested in the roles of these two factors in entropy calculation.

For each memorization score, we report the sample space size formed in Fig. 5. The statistics are calculated following the same experimental setup on OLMo-2-1124-7B. Specifically, we depict $(e, \mathcal{T}_e)$ pairs using green cross sign $\times$. The experimental result indicates that the sample space, or unique token count, grows exponentially as memorization score increases. Remarkably, while the full dataset contains tens of millions of tokens, perfect memorization (score=0) occurs within merely $2^8$ unique tokens.

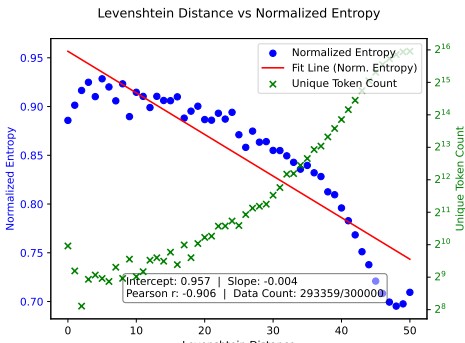

Figure 5: Memorization score *v.s.* Normalized entropy on OLMo-2-1124-7B. Note: "unique token count" scale is *exponential*.

After obtaining the sample space size, we examine how empirical probabilities are distributed in the space. It is characterized by *normalized* entropy:

$$\overline{M}(s_e) \triangleq \frac{M(s_e)}{H_{\max,e}} = \frac{M(s_e)}{\log |\mathcal{T}_e|}, \qquad (6)$$

where $\overline{M}(s_e)$ normalizes the entropy estimate $M(s_e)$ by its theoretical maximum $H_{\max,e}$. Values approaching 1 indicate a near-uniform distribution over the token set $\mathcal{T}_e$, while lower values suggest greater non-uniformity.

Following the same setup on OLMo-2-1124-7B, we plot $(e, \overline{M}(s_e))$ using blue dots on Figure 5 to observe how the normalized entropy estimator changes with the memorization score. Interestingly, we observe a linear trend that normalized entropy decreases as the memorization score increases. In appendix B.4, we discuss the pattern with different sequence length.

> **Summary.** Lower memorization-score data comprises *exponentially-linear* fewer unique tokens, and achieves *linearly* higher entropy values given the support size.

## 5.2 ENTROPY-MEMORIZATION LAW IS ROBUST UNDER DISPARATE SEMANTIC DATA

We employed a semantic-agonistic strategy to sample the dataset in the main body. This section then explores Entropy–Memorization Law under different semantic data. We chunk the sampled dataset into $k = 16$ semantic clusters, develop a strategy to find the semantics of each cluster, and then examine EM Law under these disparate semantic data. This experiment was based on the OLMo-1B model using 240,000 sample pieces.

**Semantic Clustering Pipeline** The specific steps are as follows:

1. Extracting semantics of token sequences using sentence embeddings. In this step, sentence embeddings projects a token sequence to a high-dimensional vector space, where semantically similar sequences are mapped to nearby points. Such embedding techniques are implemented by a twisted verison of pre-trained LLM.

2. Clustering. With semantic embeddings, we apply K-Means (Lloyd, 1982) in the latent space and partition the data into $k = 16$ semantic clusters.

3. Identifying semantics of the cluster. Since the clustering methods are performed in a latent space which is not interpretable, we develop an highly-automated pipeline to identify the semantics of each cluster. The core algorithm is differential clustering (Zhang et al., 2025).

4. Run Algorithm 2 on 16 partitions of the dataset. For each cluster, a linear regression is applied. We report the Pearson correlation coefficient, slope, and intercept and visualize the fitted lines.

To implement step 1, we select a popular model *all-mpnet-base-v2* (huggingface, 2025) from Sentence Transformers in Huggingface as the encoder. In Appendix C, we present how we implement step 3 in the pipeline, and provide detailed clustering results with labeled semantics.

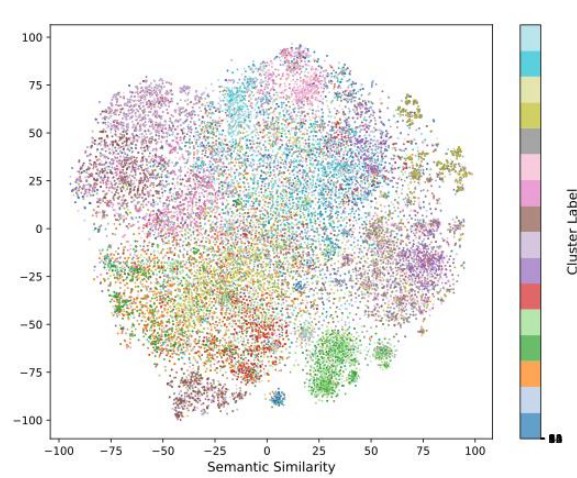

| Cluster | r | Slope | Intercept |
|---|---|---|---|
| 01 – Scientific Research and Technology | 0.82 | 0.04 | 7.86 |
| 02 – Address and Organizational Identifiers | 0.86 | 0.05 | 4.95 |
| 03 – Religious and Biblical Text | 0.91 | 0.05 | 6.56 |
| 04 – Community Engagement and Activities | 0.88 | 0.03 | 8.73 |
| 05 – Social Development and Policy | 0.88 | 0.06 | 7.27 |
| 06 – Software Development Configuration | 0.91 | 0.06 | 6.30 |
| 07 – Spiritual and Religious Beliefs | 0.91 | 0.05 | 6.87 |
| 08 – Health Risks and Medical Studies | 0.89 | 0.08 | 5.77 |
| 09 – Economic Indicators and Growth | 0.86 | 0.08 | 6.23 |
| 10 – Evolutionary Biology and History | 0.86 | 0.03 | 8.45 |
| 11 – Fantasy Narrative (Harry Potter) | 0.83 | 0.05 | 7.25 |
| 12 – Personal Experiences and Emotions | 0.89 | 0.05 | 7.22 |
| 13 – Linguistic and Semantic Analysis | 0.82 | 0.04 | 8.42 |
| 14 – Genetic and Biomedical Research | 0.75 | 0.05 | 7.39 |
| 15 – Programming Command Scripts | 0.87 | 0.02 | 8.93 |
| 16 – Political Events and Commentary | 0.89 | 0.05 | 7.53 |

Figure 6: Clustering sentence embeddings using OLMo-1B pre-training dataset. We apply T-SNE (van der Maaten & Hinton, 2008) for dimension reduction.

**Experimental Results** Figure 6 presents overall results. It is verified that Entropy-Memorization Law is observed among all clusters of data.

Moreover, another interesting finding is that, in general, different clusters exhibit distinctive intercept and slope values. For example, cluster 1 (Address and Organizational Identifiers) exhibits low intercept, while cluster 3 (Community Engagement and Activities) and 14 (Programming) exhibit high intercepts.

> We confirm that Entropy-Memorization Law is robust under disparate semantic data clusters. Moreover, intercepts and slopes are different for different semantic data.

## 6 RELATED WORK

### 6.1 FACTORS SHAPING MEMORIZATION

Since the discovery of the memorization phenomenon in the late 2010s (Zhang et al., 2017; Carlini et al., 2019; 2020; Feldman, 2020), the AI Security and Privacy research community has maintained a strong interest in the phenomenon and its implications. The following paragraphs examine how memorization in language models is influenced by key factors, including *training data, model paradigm, and prompting strategy*.

*Data shapes memorization.* Several literature suggests that (Kandpal et al., 2022; Biderman et al., 2023b) duplicated data significantly increases memorization. Larger models trained on bigger datasets show increased memorization (Biderman et al., 2023a;b). Other studies (Tirumala et al., 2022; Wang et al., 2025) investigate how memorization manifests across data with varying semantics and sources.

*Model Paradigm shapes memorization.* Beyond pre-trained language models, recent work has explored memorization in post-training stages. Chu et al. (2025) demonstrate that supervised fine-tuned (SFT) LLMs exhibit stronger memorization tendencies than those trained with reinforcement learning (RL). Additionally, Nasr et al. (2025) reveal that safety-aligned models still retain memorized data.

*Prompting Strategy shapes memorization.* Researchers employ three main types of prompting strategies for language models, categorized by threat models. A significant body of work relies on manual efforts or template-based approaches to generate prompts at scale, as seen in Carlini et al. (2019; 2020); Kim et al. (2023). Studies such as Carlini et al. (2020); Kandpal et al. (2022); McCoy

et al. (2021) demonstrate that longer prompts substantially increase the likelihood of reproducing memorized training data sequences. Another line of research constructs prompts directly from existing data sources, such as training corpora or web data (Nasr et al., 2025; Carlini et al., 2023; Kandpal et al., 2022; Ippolito et al., 2023; Aerni et al., 2025). Recent advances involve more sophisticated strategies that leverage synergies between LLMs and training data. For instance, Zhang et al. (2023) quantify how a model's performance on an example $x$ depends on whether $x$ was included in the training data. Additionally, Schwarzschild et al. (2024) adapts GCG (Zou et al., 2023)—a prompt optimization tool originally designed for adversarial attacks—to generate effective extraction prompts.

## 7 LIMITATIONS AND BROADER IMPACTS

### 7.1 LIMITATIONS

*Empirical experiments.* Our work adopts a single set of prompting strategy (DM) and memorization score (edit distance) in our memorization experiments. Although the setup is commonly used by other studies, other combinations exist. We would like to explore adversarial compression (Schwarzschild et al., 2024), and non-adversarial reproduction (Aerni et al., 2025) in our future work.

*Memorization score prediction.* Due to limitation of sample space as discussed in Section 3, our strategy does not enable memorization score prediction at the instance level.

### 7.2 IMPLICATIONS AND SOCIETAL IMPACT

EM Law facilitates the theoretical understanding of factors in LLM memorization. For trainers of LLMs, our work contributes a guideline in model audition: lower entropy data is at higher risk of leakage. LLMs trainers can pre-screen training datasets to assess memorization risk.

Besides, by applying Entropy-Memorization Law on test data, it is observed that the plot behaves very differently from training dataset. we discover a simple strategy for the Dataset Inference (DI) task. DI aims to tell the membership (1 for training data, and 0 for testing data) at dataset-level. We defer the details to Appendix D.

To the best of the authors' knowledge, this research does not introduce any additional negative societal impacts.

## 8 CONCLUSIONS

This paper presents Entropy-Memorization Law: a level-set-based entropy estimator of training data chunks linearly approximate edit-distance-based memorization score. Further investigation indicates that in EM Law is robust under different sequence length, sampling strategies, and data clusters with different semantics. By examining vocabulary size, it is revealed that lower memorization-score data comprises *exponentially-linear* fewer unique tokens, and achieves *linearly* higher entropy values given the support size.

For future work, we plan to explore why the proposed level-set-based entropy estimator fits memorization score so well. Potential theoretical tools include the long-tail theory by Feldman and other researchers (Feldman, 2020; Feldman & Zhang, 2020), and multi-calibration in LLMs (Detommaso et al., 2024). Such efforts may also shed light on interpreting slope and intercept resulting from the EM Law.

## REPRODUCIBILITY STATEMENT

This paper uses existing open-research LLMs and their corresponding training datasets. All model weights, training datasets are accessible through web hosting services. The detailed process on data preparation has been included in Section 2. For algorithms adopted in this paper, zlib is free to access online; and we have provide enough details for the level-set-based entropy estimator.

After the submission is accepted, we will release related code to the research community.

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

## A  DETAILS ON EXPERIMENTAL SETUP

### A.1  COMPOSITION OF SAMPLED DATASET

We constructed datasets with sizes 240,000 and 300,000, respectively, for Dolma and OLMo-2-1124-Mix. Table 1 and 2 presents the composition of sampled datasets after LCS filtering.

## B  EXTENDED RESULTS ON ENTROPY-MEMORIZATION LAW

### B.1  INSTANCE-LEVEL COMPRESSIBILITY

Figure 7 presents the result of instance-level zlib compression ratio. It fails to fit memorization score.

### B.2  ADDITIONAL RESULTS WITH VARIOUS SAMPLING STRATEGY OF LLMS

In the main body of the paper, we assume a fixed temperature of 0.8. In this subsection, we adopt different sampling strategies of LLMs and discuss how these strategies might shape EM Law. Due to computation constraints, we conduct our experiments on a subset "DCLM1" with OLMo-2-1124-7B. The size of the subset is around 28,000.

Table 2: Source Counts of OLMo-2-1124-Mix

| Dataset | Count |
|---|---|
| algebraic-stack | 797 |
| arxiv | 1,464 |
| dclm1 | 28,130 |
| dclm2 | 27,984 |
| dclm3 | 28,014 |
| dclm4 | 28,163 |
| dclm5 | 28,049 |
| dclm6 | 28,065 |
| dclm7 | 28,060 |
| dclm8 | 28,091 |
| dclm9 | 28,040 |
| dclm10 | 28,032 |
| open-web-math | 499 |
| pes2o | 4,364 |
| starcoder | 5,280 |
| wiki | 276 |
| Total | 293,308 |

Table 1: Source Counts of Dolma

| Source Dataset | Count |
|---|---|
| pes2o | 39,777 |
| cc | 39,743 |
| books | 39,598 |
| reddit | 39,373 |
| stack | 38,865 |
| wiki | 23,115 |
| Total | 219,186 |

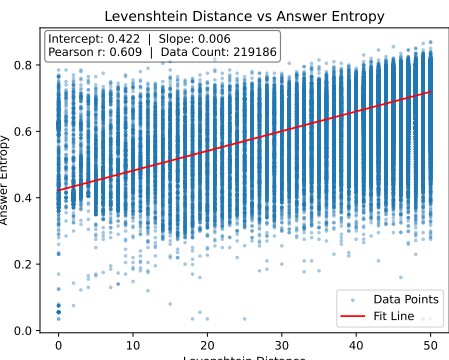

Figure 7: Instance-level zlib compression rate v.s. memorization score on Olmo-1B

We consider combinations of temperature, top-k sampling, and nucleus sampling (top-p). The experimental results are summarized in Tab. 3, and details are shown in Fig. 8 - 13. Under all sampling strategies we have explored, we empirically observe that EM Law holds with $r > 0.92$. Beyond that, we made a few observations here:

- The zero-distance point $(0, M(s_0))$ exhibits a significant deviation from the regression line in both plots.

- Intercept and slope are dependent if we fix the LLM and dataset. The general pattern is that when the intercept increases, the slope decreases. In fact, when the memorization score e= 50. Besides, $M(s_{50}) \approx 11$. Although different regression lines have different slopes and intercepts, the y-value reaches about 11. This might indicate that intercept and slope may have a degree of freedom 1.

- With a fixed temperature, enabling top-k or top-p sampling increases intercept and decreases slope.

- The estimated normalized entropy decreases with the memorization score increasing.

The first two observations are consistent with observation points 2 and 3 in Section 5.

Table 3: Entropy-Memorization Law under different LLM sampling strategy

| Strategy | r | Intercept | Slope |
|---|---|---|---|
| Temp=0 | 0.933 | 5.490 | 0.106 |
| Temp=0.5 | 0.936 | 5.474 | 0.106 |
| Temp=0.8 | 0.926 | 5.011 | 0.113 |
| Temp=0.8, top_p=0.5 | 0.935 | 5.599 | 0.103 |
| Temp=1 | 0.944 | 4.646 | 0.118 |
| temp=0.8, top_k=10 | 0.944 | 5.138 | 0.111 |

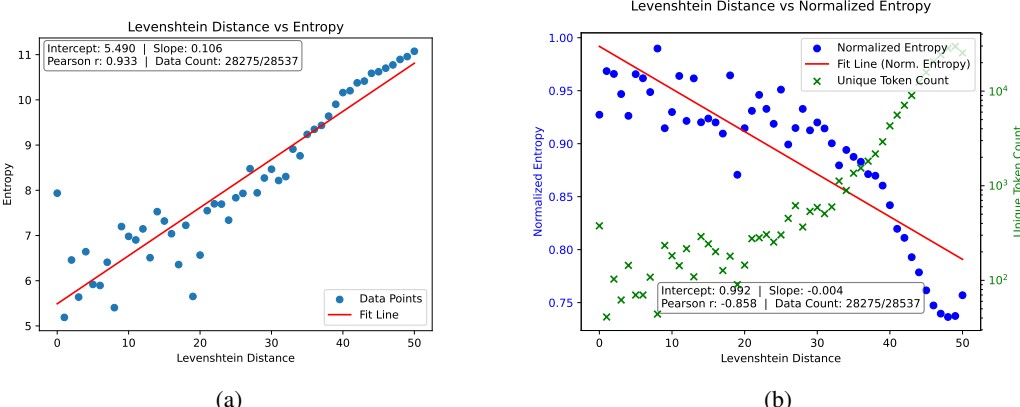

(a)               (b)

Figure 8: Temp=0

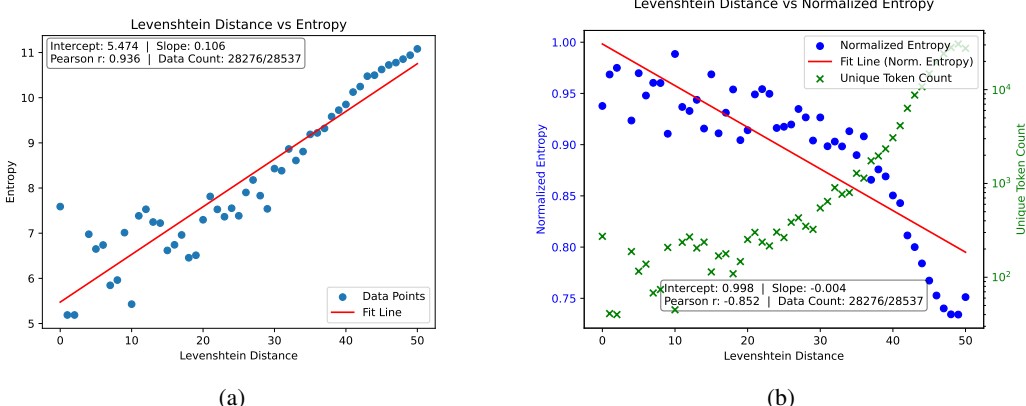

(a)               (b)

Figure 9: Temp=0.5

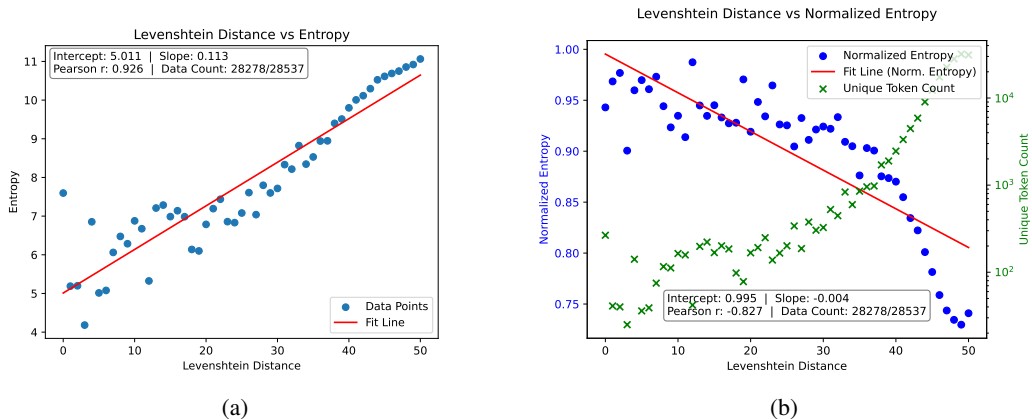

(a)                                                      (b)

Figure 10: Temp=0.8

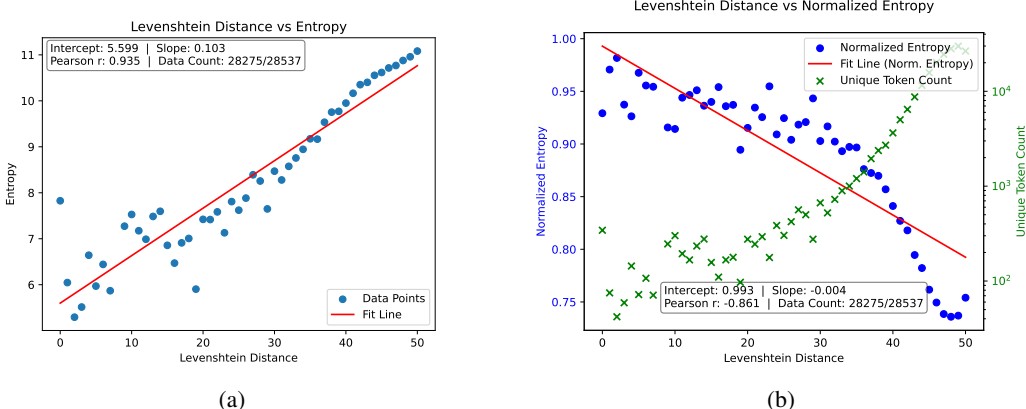

(a)                                                      (b)

Figure 11: Temp=0.8, Top p=0.5

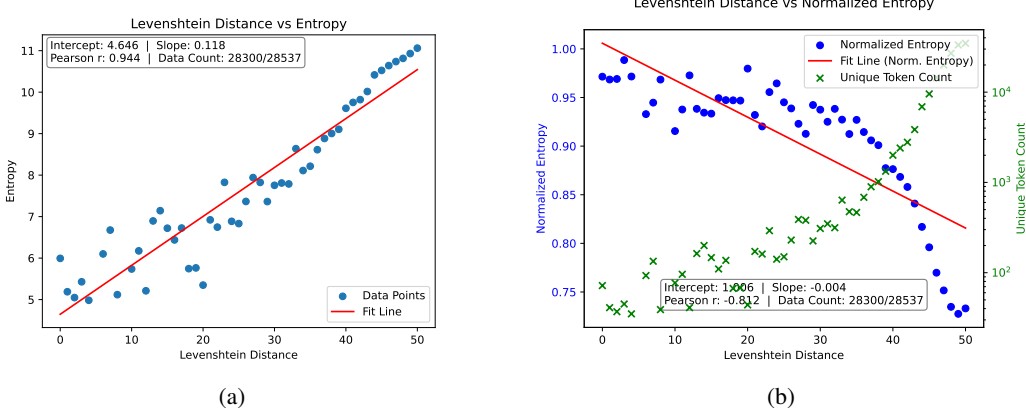

(a)                                                      (b)

Figure 12: Temp=1

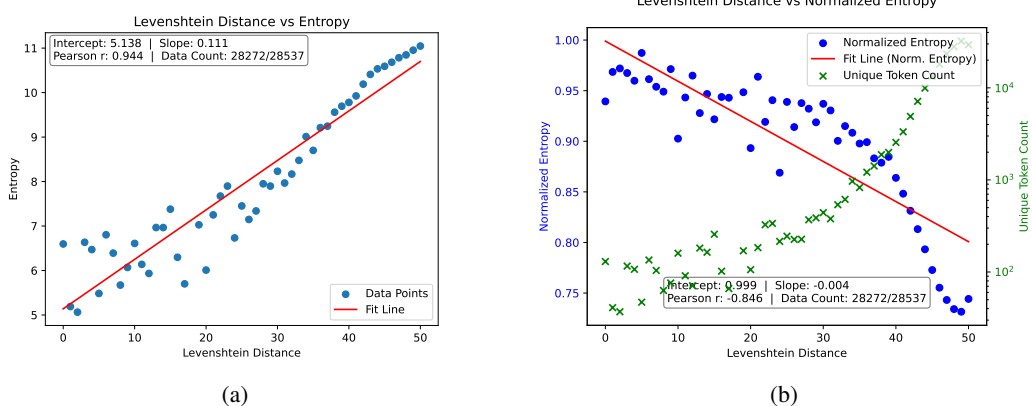

Figure 13: Temp=0.8, top-k=10

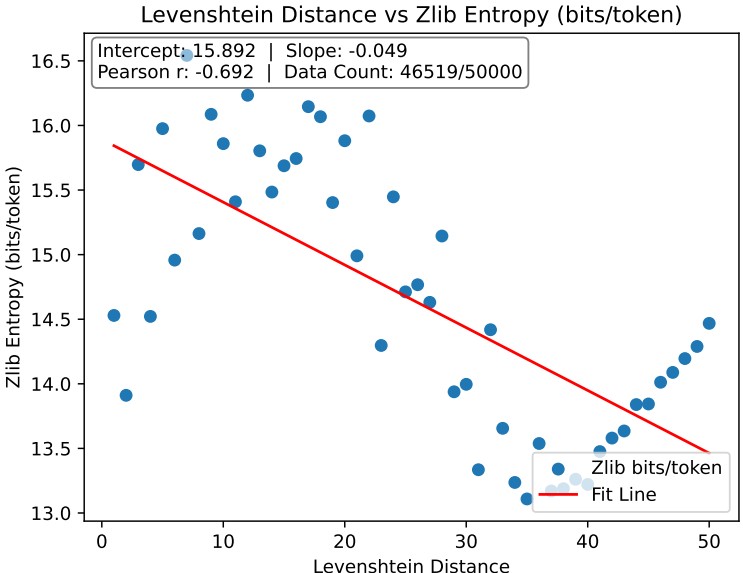

Figure 14: zlib compression rate v.s. memorization score for OpenLlama-7B

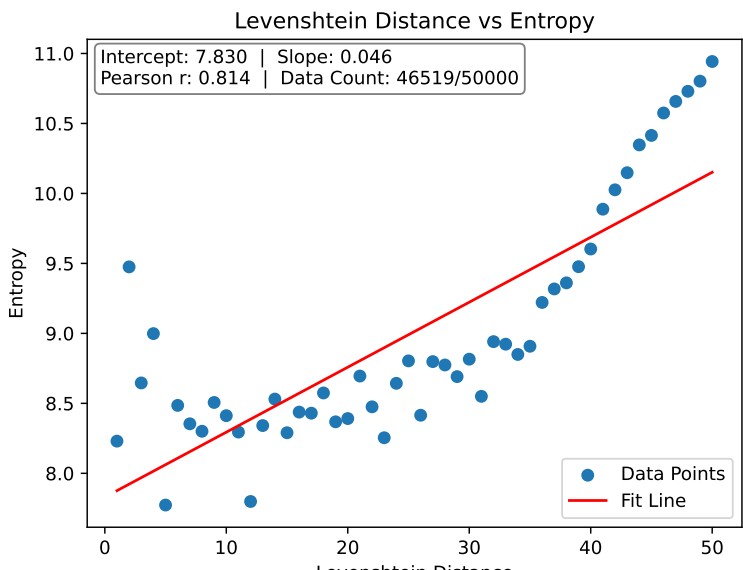

Figure 15: Level-set-based Entropy estimate v.s. memorization score for OpenLlama-7B

### B.3 ENTROPY-MEMORIZATION LAW

The experimental results on OpenLlama-7B are shown in Figure 14 and 15. The findings are consistent with the pattern observed in the main paper body.

### B.4 NORMALIZED ENTROPY–MEMORIZATION LAW

Figure 16: Normalized Entropy–Memorization Law for OpenLlama-7B

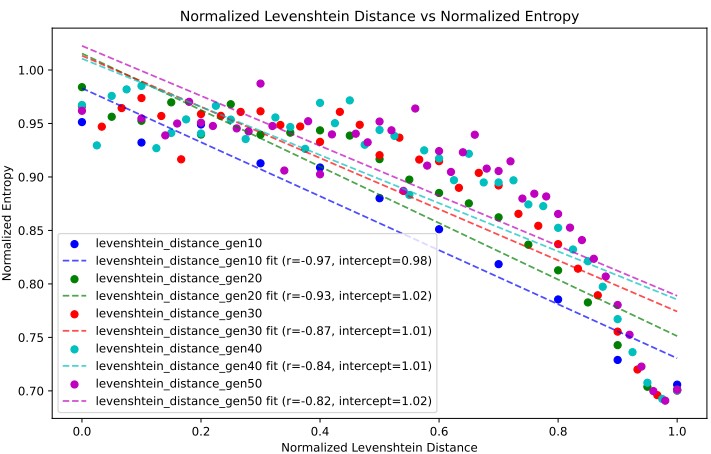

Figure 17: Estimated normalized entropy vs memorization score.

Fig. 17 presents the (estimated) normalized entropy *v.s.* memorization score. The findings are consistent with the findings that we observed in main paper body. Moreover, $|r|$ increases from 0.82 to 0.97 as generation length decreases.

## C    EXTENDED RESULTS ON ENTROPY-MEMORIZATION LAW UNDER DISPARATE SEMANTIC DATA

### C.1    TECHNICAL DETAILS ON INTERPRETING SEMANTICS OF EACH CLUSTER

To identify the semantics of each cluster, we build a pipeline that significantly reduces human annotation efforts. The pipeline is as follows:

1. Detect distinctive samples within each cluster. Zhang et al. (2025) formulates this task as a *differential clustering* problem and proposes a FINC method. To quantitatively measure semantic distinctions among the 16 clusters obtained via K-means, we conducted 16 FINC comparisons. For each cluster $C_i$, we set $C_i$ as the novel dataset and the union of the remaining 15 clusters as the reference set. The input to FINC is the sentence embeddings of all instances in the set, and FINC suggests the distinctive samples in the cluster.

2. Keywords summarization. In this stage, we use tri-grams as effective descriptors for naming and interpreting cluster identities. Specifically, we use i) *spaCy* (PyPI, 2025) to perform named entity recognition and dependency parsing to ensure that extracted units are linguistically complete phrases (e.g., *"protective spell harry"*, *"lend broom fly"*), and ii) *YAKE* (Campos et al., 2020) to ranks terms using heuristics such as frequency, context, and positional distribution.

3. Human annotation. Based on summarized keywords, human annotators further summarize the semantics of the cluster.

### C.2    ENTROPY–MEMORIZATION PLOT FOR EACH CLUSTER

### C.3    INTERPRETING SEMANTICS OF EACH CLUSTER

Table 4 presents top-5 keywords and human-annotated semantic labels for each cluster.

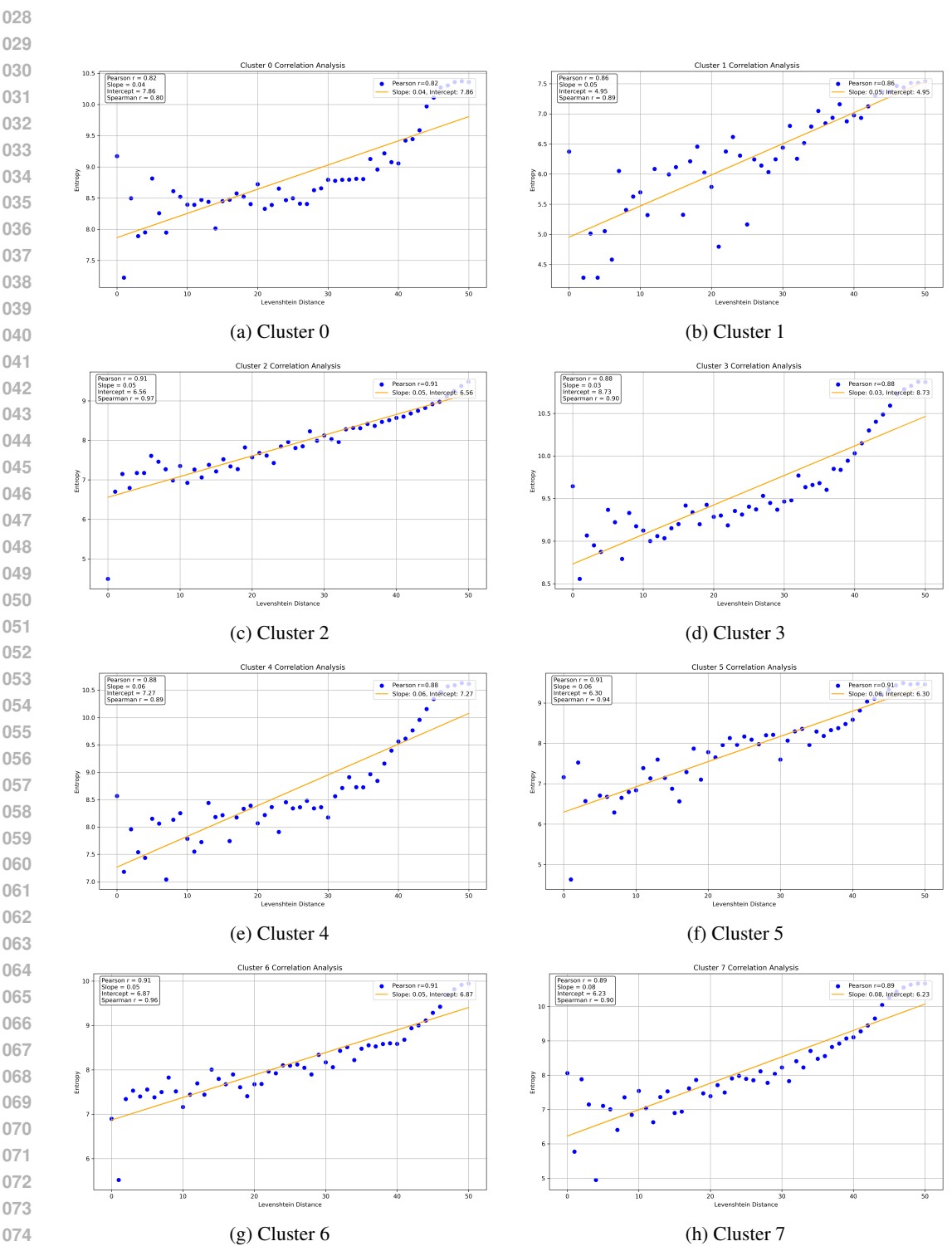

Figure 18: Clusters 0–7.

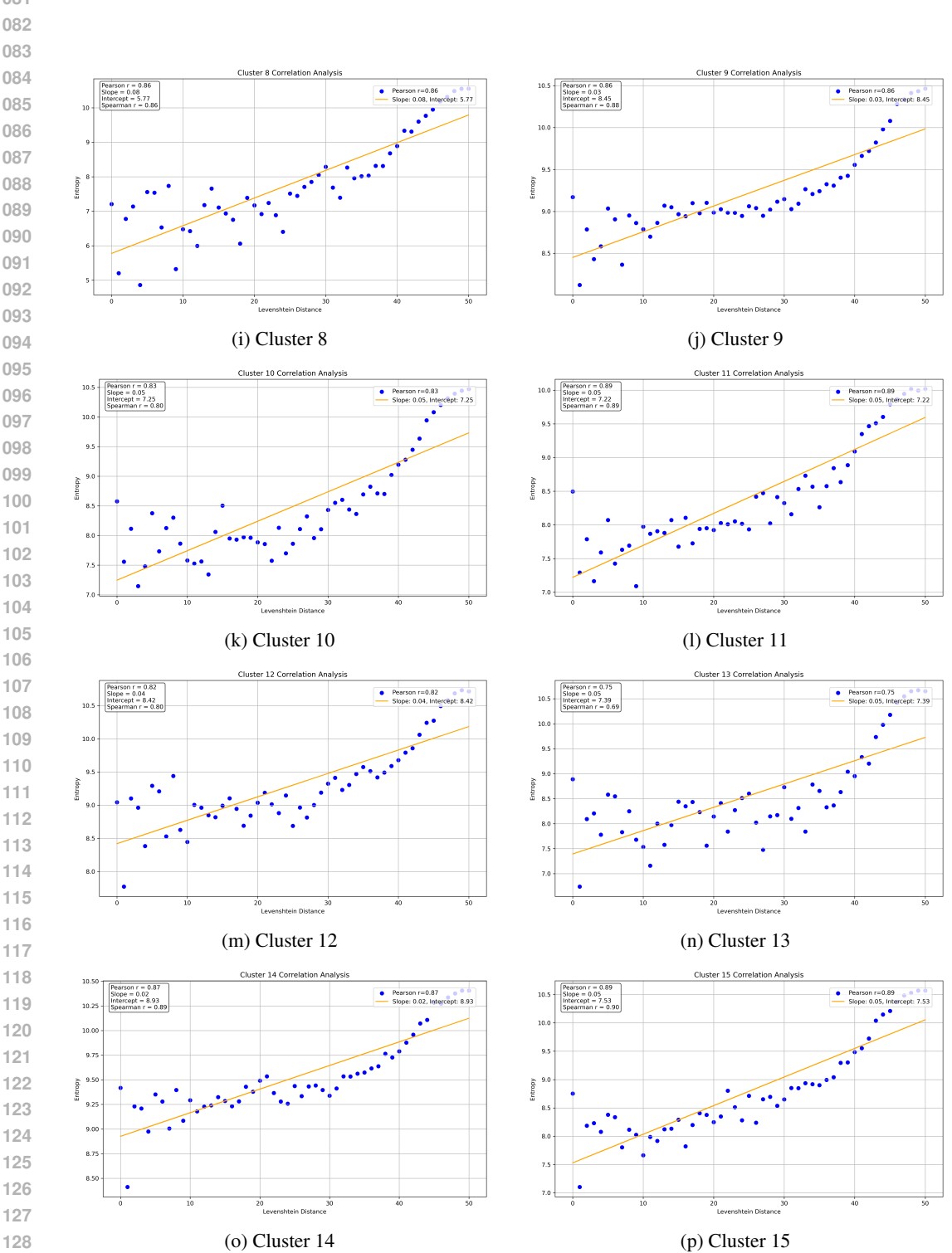

Figure 18: Clusters 8–15.

Table 4: Semantics of each cluster (cluster 0-7)

*Cluster 0: Scientific Research and Technology*

| Score | Keyword Phrase |
|---|---|
| 1.28e-04 | translate depth direction |
| 1.28e-04 | design fabrication characterization |
| 1.28e-04 | pct design fabrication |
| 1.28e-04 | manipulation pct design |
| 1.23e-04 | sensor control manipulation |

*Cluster 1: Address and Organizational Identifiers*

| Score | Keyword Phrase |
|---|---|
| 1.74e-05 | street number city |
| 1.54e-05 | party committee number |
| 1.54e-05 | city number |
| 1.52e-05 | type number form |
| 1.46e-05 | conduit state number |

*Cluster 2: Religious and Biblical Texts*

| Score | Keyword Phrase |
|---|---|
| 7.94e-05 | people sword thy |
| 7.94e-05 | thy people sword |
| 7.91e-05 | jacob son reuban |
| 7.84e-05 | son brother house |
| 7.80e-05 | son lord hath |

*Cluster 3: Community Engagement and Activities*

| Score | Keyword Phrase |
|---|---|
| 1.05e-04 | irrigation evaporate leave |
| 1.05e-04 | bullet time jump |
| 1.04e-04 | community meetup world |
| 1.03e-04 | community kid spout |
| 1.02e-04 | year electronic music |

*Cluster 4: Social Development and Policy*

| Score | Keyword Phrase |
|---|---|
| 1.34e-04 | dramatically drive price |
| 1.33e-04 | develop skill team |
| 1.33e-04 | prefer policy influence |
| 1.33e-04 | outwith prefer policy |
| 1.33e-04 | sector real passion |

*Cluster 5: Software Development Configuration*

| Score | Keyword Phrase |
|---|---|
| 2.39e-05 | true plugin proposal |
| 2.29e-05 | node optional true |
| 2.27e-05 | header content type |
| 2.26e-05 | true ellipsis true |
| 2.26e-05 | dev true child |

*Cluster 6: Spiritual and Religious Beliefs*

| Score | Keyword Phrase |
|---|---|
| 9.67e-05 | thy mind thy |
| 9.61e-05 | death eternal life |
| 9.54e-05 | create sustain universe |
| 9.52e-05 | world drive ulterior |
| 9.52e-05 | behavior world drive |

*Cluster 7: Health Risks and Medical Studies*

| Score | Keyword Phrase |
|---|---|
| 1.06e-04 | health increase risk |
| 1.06e-04 | evaluate risk factor |
| 1.05e-04 | pregnancy increase risk |
| 1.05e-04 | disease high risk |
| 1.05e-04 | high disease risk |

Table 5: Semantics of each cluster (cluster 8-15)

*Cluster 8: Economic Indicators and Growth*

| Score | Keyword Phrase |
|---|---|
| 2.54e-05 | billion country oda |
| 2.51e-05 | permanent surface runway |
| 2.51e-05 | kwh capita growth |
| 2.50e-05 | imf intelsat interpol |
| 2.48e-05 | price growth rate |

*Cluster 9: Evolutionary Biology and History*

| Score | Keyword Phrase |
|---|---|
| 4.49e-04 | primatology million year |
| 4.48e-04 | mutate gene ancient |
| 4.48e-04 | start million million |
| 4.45e-04 | paleolithic million year |
| 4.44e-04 | account million year |

*Cluster 10: Fantasy Narrative (Harry Potter)*

| Score | Keyword Phrase |
|---|---|
| 4.98e-05 | dumbledore harry meet |
| 4.95e-05 | forward harry ron |
| 4.94e-05 | muggle bystander incredulously |
| 4.72e-05 | sirius black peril |
| 4.71e-05 | harbor end time |

*Cluster 11: Personal Experiences and Emotions*

| Score | Keyword Phrase |
|---|---|
| 1.07e-04 | hunt season roll |
| 1.06e-04 | love good love |
| 1.06e-04 | thing work happen |
| 1.05e-04 | decision normal result |
| 1.04e-04 | understand thing happen |

*Cluster 12: Linguistic and Semantic Analysis*

| Score | Keyword Phrase |
|---|---|
| 1.22e-04 | interpret apply male |
| 1.20e-04 | ingen det finne |
| 1.20e-04 | finne ingen det |
| 1.19e-04 | title aktivt medlem |
| 1.17e-04 | confusion attain agenda |

*Cluster 13: Genetic and Biomedical Research*

| Score | Keyword Phrase |
|---|---|
| 8.51e-05 | mesenchymal stem cell |
| 8.40e-05 | screen gene foster |
| 8.27e-05 | interaction show high |
| 8.26e-05 | expression level gene |
| 8.25e-05 | search perform blast |

*Cluster 14: Programming Command Scripts*

| Score | Keyword Phrase |
|---|---|
| 4.64e-06 | text text ohm |
| 4.57e-06 | delaytimer command |
| 4.03e-06 | dark text text |
| 3.94e-06 | command runcmd |
| 3.66e-06 | cost command type |

*Cluster 15: Political Events and Commentary*

| Score | Keyword Phrase |
|---|---|
| 1.41e-04 | white house official |
| 1.41e-04 | idea african americans |
| 1.38e-04 | african americans woman |
| 1.38e-04 | end war year |
| 1.37e-04 | snc lavalin scandal |

# D  DATASET INFERENCE USING EM LAW

## D.1  ENTROPY MEMORIZATION LAW ON TEST DATA

Running algorithm 2 on *training* dataset of LLMs, we discovered Entropy-Memorization Law. This section then explores another question that naturally arises – what happens if we run the same algorithm on *test* dataset? It turns out that the plot behaves very differently from training datasets.

Figure 19 presents the plot of applying the entropy estimator and normalized entropy estimator to test data. To maintain consistency throughout the paper, we use the terms used for memorization with slight abuse. For example, the memorization score still measures the distance between the ground truth and the model's response, but the model is not actually "memorizing" training data. Following the same setup described in the main body, we select data from LiveBench (White et al., 2025) from 2024-06-25 to 2024-11-25. The time property guarantees that LiveBench is non-member data for OLMo-2-1124-7B. LiveBench is licensed under CC BY-SA 4.0 International License.

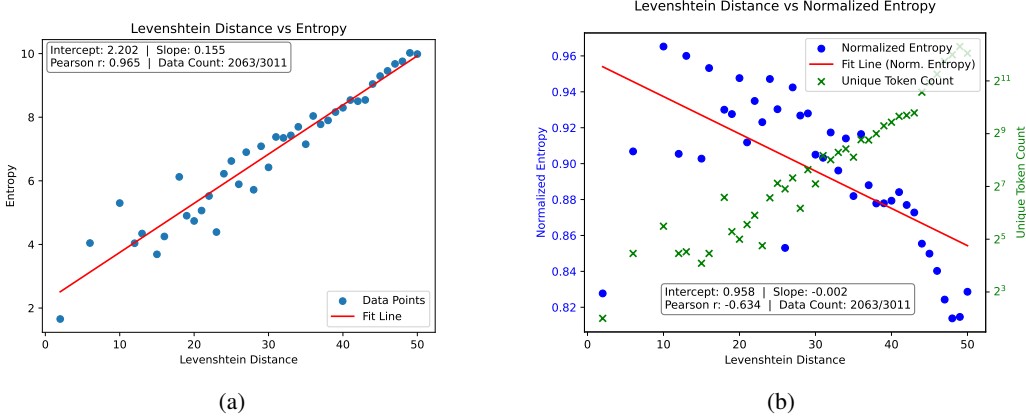

(a)                                                      (b)

Figure 19: Entropy Memorization Law on OLMo-2 with LiveBench dataset.

With around 3k samples from LiveBench, we observe that:

- Level-set-based entropy is still a good indicator of LLM performance, as demonstrated by the high Person's $r$. The sample space also approximately grows exponentially as the memorization score increases.

- However, compared with training data, the intercept is lower, and the slope is higher. There is less coverage of low memorization scores, especially for memorization scores within 0-10.

- at low-distance set, it is observed that OLMo-2 is producing low-entropy text. The entropy is much lower than what we observed on train data.

Here is a case study on a 2-distance memorization:

*Prompt Text*

Please create a valid join mapping between CSV Table A and CSV Table B. Each column in A maps to 0 or 1 columns in B. Return your response as a Python dictionary, formatted as col_name_in_df_a : col_name_in_df_b. Please return only the dictionary.
CSV Table A: Areas,freq_1,freq_2,freq_3,freq_4,freq_5,freq_6 0.0,0.0,0

*Generated Text*

.0,0.0,0.0,0.0,0.0
1.0,0.0,0.0,0.0,0.0,0.0,0.0,0.0
2.0

Given the prompt given in this case study, OLMo-2 generates repeated numbers, exhibiting low entropy. Based on the above observations, we thus derive a simple strategy to tell the difference between train data and test data for LLM. In the AI community, this task is studied under the name *Dataset Inference*.

## D.2 EMBEDI: ENTROPY-MEMORIZATION LAW-BASED DATASET INFERENCE

**Dataset Inferences** Dataset Inference (DI) (Maini et al., 2024; 2021) builds on the idea of membership inference attacks (MIA). While both MIA and DI aim to identify whether some suspect data was part of the training data, they differ on the amount data required. MIA operates at *instance* (sentence) level, however, DI operates on a *collection* of instances – in reality, the suspect data used for DI could be a book.

In practice, DI can identify potential test set contamination, to provide a calibrated performance evaluation of LLMs. DI may also detect unauthorized usage of copyrighted training data, thus promoting the protection of intellectual properties. Moreover, by using more data to determine membership, DI is deemed to be more realistic than MIAs. As Maini et al. (2021) presented that as the size of the training set increases, the success of membership inference degrades to random chance. EMBEDI is inspired by several empirical observations. First, over-parametrization of LLMs may lead to overfitting on training data, resulting in a generalization gap between training data and testing data; then, LLM may perform well on low-entropy testing data, resulting in a low intercept in EM Law. Besides, given a fixed dataset and LLM, empirical evidence suggests that the intercept and slope generated by Algorithm 2 are dependent. This inspires us to develop the following strategy for dataset inference:

> Given an LLM $\theta$ and dataset $D$, run Algorithm 2 and get intercept $k$. Compare $k$ with a pre-defined threshold $\tau_k$. Assign label 1 (i.e., member) if $k > \tau_k$, assign label 0 otherwise.

**Amount of data required for EMBEDI** Effective dataset inference requires reliable entropy estimation and diverse memorization score distributions. In the main body, we have revealed that the frequency of a low memorization score is exponentially smaller than that of a high memorization one. We, therefore, set the minimum sample size to $n = 1,500$, with each sample is a 150-token sequence.

## D.3 EXPERIMENTAL RESULTS ON DATASET INFERENCE

Table 6: Extended result on dataset inference

| LLM | Dataset | Intercept | Slope | Prediction | Label |
|------|----------------|-----------|--------|------------|-------|
| OLMo-2 | LiveBench | 2.202 | 0.155 | 0 | 0 |
| Pythia | MIMIR_cc | -2.048 | 0.251 | 0 | 0 |
| Pythia | MIMIR_cc | 3.992 | 0.091 | 1 | 1 |
| OLMo-2 | OLMo-2-1124-Mix | 3.724 | 0.142 | 1 | 1 |
| Pythia | MIMIR_full | 6.297 | 0.092 | 1 | 0 |
| Pythia | MIMIR_full | 6.166 | 0.095 | 1 | 1 |
| Pythia | MIMIR_tarxiv | 1.156 | 0.174 | 1 | 1 |
| Pythia | MIMIR_tarxiv | -0.910 | 0.0227 | 0 | 0 |
| Pythia | MIMIR_wiki | 3.006 | 0.131 | 1 | 0 |
| Pythia | MIMIR_wiki | 2.894 | 0.133 | 1 | 1 |

*Selected LLMs.* We select OLMo-2-1124-7B ("OLMo-2"), Pythia-6.9b-deduped (Biderman et al., 2023b) ("Pythia").

*Selected Datasets.* We select LiveBench (White et al., 2025) and MIMIR (Duan et al., 2024). For LiveBench, we use data from 2024-06-25 to 2024-11-25. MIMIR is a public dataset originally for evaluating MIAs on the Pythia suite by re-compiling the Pile (Gao et al., 2020) train/test splits. For MIMIR, we use "Pile CC" "temporal arXiv", "wiki" subset, and full dataset ("full") for evaluation.

Note that LiveBench and Temporal arxiv are temporal-cutoff-based, while the remaining dataset is i.i.d.-based.

*Threshold for each LLM.* In EMBEDI, we assign a threshold for each LLM. We empirically set $\tau_k$ to 0 and 3 for Pythia and OLMo-2, respectively. Other variants of thresholding strategy include domain-specific thresholding, and we leave it to future work.

Table 6 presents the overall results of EMBEDI on the dataset inference task.

**Discussions** EMBEDI enjoys several advantages: it is compute-efficient – it only requires LLM inference on $n$ samples. No training is required. It does not require any additional reference models.

# E COMPUTE

All experiments were conducted on a GPU cluster equipped with 4 NVIDIA RTX 3090 GPUs (24GB CUDA memory per card), running Ubuntu 22.04. We use VLLM (Kwon et al., 2023) to speed up inference.

