# OpenReview forum: "Entropy Proxy for LLM Memorization Score"
_ICLR.cc/2026/Conference — ICLR 2026 Conference Withdrawn Submission_

### Official Review · Reviewer_BafX · 2025-10-19

**Soundness:** 2
**Presentation:** 1
**Contribution:** 3
**Rating:** 2
**Confidence:** 3

**Summary:**

The main question of the paper is to quantitatively characterize memorization difficulty of training data by an LLM using the intrinsic property of the data itself, such as entropy. The paper focuses on approximating memorization score, according to discoverable memorization, using the entropy of a set of instances. The key contribution is the entropy-memorization law, which suggests that data entropy is linearly correlated with memorization score. The law is derived based on empirical observation of entropy vs edit distance between ground-truth generation and LLM response.

**Strengths:**

The key strength of the paper is the connection between data entropy to memorization score incurred by an LLM. The observation that lower entropy data is at higher risk of memorization leakage has implications in future experiments. Experimental results look convincing -- I like the part where the authors realize that instance-level entropy estimate becomes noisy and requires further analysis, such as extending to set-level analysis.

Overall, the paper is potential to quantify the difficulty of memorization from the aspect of training data.

**Weaknesses:**

The paper lacks from preciseness. I elaborate below.

- There is no formal statement of entropy-memorization law. Rather, the readers have to understand the law from Figure 5, which has an additional axis of unique token counts, making it difficult to get the main point.
- Some choices look arbitrary and not well justified. For example, edit distance only looks into syntactic similarity, but does not consider the intrinsic difficulty of generating a response. A response with very low generation probability by the LLM may have lower edit distance with the ground-truth and therefore, tagged as memorized -- essentially, the paper neglects the intrinsic difficult of token generation by the LLM and its connection to memorization.
- The choice of not including post-trained LLMs looks unjustified. Anyway, the proposed methodology is agnostic to training strategy.
- I do not get why filtering trivial memorization is needed. There could be cases when the response needs to contain part of the prompt. Why the authors think that memorization is unlikely in those scenarios?
- The authors find that instance-level entropy cannot explain memorization well. However, memorization is usually concerned at the level of individual instance. Moreover, the relation with sample space being smaller or large does not make sense to me. Is the analysis conducted for a single instance? If there are multiple instances, I do not understand why sample space range would be upper bounded by 50?
- In Equation 1, independent token probability is considered. I do not know if *point probability* is a standard term. Is there a way to extend to conditional token probability, since this is how autoregressive LLMs work? Does it help with removing noise by looking at instance-level entropy? More specifically, the relation to vocabulary size has to be explained well.
- Line 77 and Line 455-456 look conflicting with each other. Which one is correct? Low entropy = high memorization, or low entropy = low memorization?
- Line 76, the readers do not know the definition of $r$.

**Questions:**

Address the points in the weakness.

---

### Official Review · Reviewer_H99e · 2025-10-26

**Soundness:** 3
**Presentation:** 2
**Contribution:** 3
**Rating:** 4
**Confidence:** 3

**Summary:**

The paper investigates how intrinsic properties of training data influence memorization in large language models. It proposes using data compressibility as a proxy to estimate how difficult a text snippet is for a model to memorize. The authors first test instance-level metrics such as zlib compression and empirical entropy, but these show weak correlation with memorization because each sequence is too short to provide a stable estimate.

They introduce a set-level approach that groups sequences by their memorization score and calculates entropy across each group’s aggregated tokens. This greatly increases sample size and produces strong linear correlations between entropy and memorization across multiple model families and training datasets. The result is presented as the Entropy–Memorization Law: text with higher entropy tends to have higher memorization difficulty.

The paper also examines the effect of token vocabulary size, sequence length, sampling strategy, and different semantic domains, and finds that the relationship remains consistent. A normalized entropy variant is presented to separate vocabulary size from token distribution uniformity. The authors argue that these findings can help assess privacy risks in training data and contribute to understanding memorization behavior in language models.

**Strengths:**

Strengths -

- The paper takes logical next steps to explore the space and runs experiments to verify their proposed law under varying conditions
- The results generalize to varying data subsets of the chosen dataset
- The paper tests across several model families and datasets corresponding to them.

**Weaknesses:**

Weaknesses -

- Does filtering data already create some bias? Will the estimator fit when you have high overlap between prompt and continuation?
- The authors claim it can be used to audit models but applying the law in practice will require computing entropy over trillions of tokens and then identifying sequences which are at risk of memorization however the paper does not talk about the effect of dilution/repetition which may be the primary cause of some sequences being memorized/not-being memorized as they are seen several times or seen at the start and never seen again so the law should account for position in training as well to actually be impactful. Repetitions are handled during entropy computation I suppose.
- The paper is overall very poorly written with grammar issues and repetition/figures are hard to read and seem rushed.
- A lot of the ideas seem to overlap with [1] especially around instance level and have been shown to not be strong predictors such as compressibility. [1] is already cited but these points are not brought up

[1] Recite, Reconstruct, Recollect: Memorization in LMs as a Multifaceted Phenomenon (https://arxiv.org/abs/2406.17746)

**Questions:**

Questions -

- Line 157 - Isn't the opposite true? If there is more randomness the compressiblity should be low.

Some Writing Nits -

- Line 83 - "A model-agnostic is compute-efficient." - Issue with writing
- Line 107 is redundant and broken
- Line 445 - Audition -> Auditing?

Relevant Citations -

- Understanding Memorisation in LLMs: Dynamics, Influencing Factors, and Implications (https://arxiv.org/abs/2407.19262) - Explores memorization with synthetic data and discusses effect of entropy on memorizability

---

### Official Review · Reviewer_jV3X · 2025-10-28

**Soundness:** 2
**Presentation:** 2
**Contribution:** 2
**Rating:** 2
**Confidence:** 3

**Summary:**

This paper studies the problem of quantitatively measuring how difficult an example is to memorize. Building upon existing notions of example compressibility, they show that a naive approach to estimating the instance-level entropy can be too noisy to adequately correlate with the memorization score. To address this shortcoming, they propose a level-set based estimator of the instance entropy. In particular, they show that this level-set based estimator correlates linearly with memorization scores  Next, they show that memorized examples comprise an exponentially smaller subset of the token space and once the entropy is normalized with respect to the total token support, the trends reverse. Finally, the authors establish that the identified entropy/memorization law is consistent across different semantic clusters of the pretraining data.

**Strengths:**

This paper attempts to solve an important and interesting problem for large language models: how easily are different sequences memorized? Although there are various intuitive heuristics present in the prior work, this paper focuses on deriving a quantitative description to predict what examples are faster to memorize versus generalize. Thus, I believe that the problem statement posed is interesting. The paper also conducts a methodological investigation, exploring various facets of their metric which increases the readers understanding of memorization in LLMs.

**Weaknesses:**

(1) Issues with clarity: I found the paper at some point difficult to understand. For example, after investigating the failures of instance-level entropy calculation, the paper introduces their level-set based estimator. However, I found this transition and the motivation of the level-set entropy calculation to be difficult to motivate. The paper notes that the level-set aggregates more samples than the instance level methodology. However, this still feels incomplete: there are many potential ways by which to aggregate the sequences to generate less noisy entropy estimates. Why are level-sets of the memorization scores the appropriate mechanism by which to aggregate. The lack of justification for this choice, and the lack of ablation of other choices feels scientifically incomplete and leaves the possibility of better estimators/proxies of memorization.

(2) Entropy Metric Seems Post-Hoc: As far as I can tell, the metric introduced here does not have predictive power. Since the entropy estimates depend on the memorization score, it does not appear possible to actually use the entropy to predict which examples are likely to be memorized. If this is indeed the case, it appears to be a significant weakness of the method: what would be the motivation for computing the entropy proxy if I need to compute the memorization score before-hand anyway. I would definitely appreciate if the authors could dispel my misunderstanding here, or explain what was their intended use-case for the entropy proxy.

(3) Insufficient Analysis/Explanation of Results: I found it quite confusing, for example, that the entropy-memorization law seems to reverse once normalized entropy is introduced. Although the authors do note this, they don't provide any explanation or hypotheses for why the trend becomes negative. It would significantly enhance the interpretability of their results and their contributions if they would provide some more interpretation of this finding. Similarly, the experiment with the semantic consistency was not entirely explained.

**Questions:**

(1) Please add some more explanation about why aggregation across the level sets was preferred rather than other aggregation techniques to denoise the entropy estimates?

(2) Please clarify whether the entropy proxy can be used to predict which sequences are likely to be memorized a-priori (without training the model). If this is not possible, please clarify what is the intended usage of this proxy.

(3) Please clarify why does the entropy-memorization law reverse after normalization (or if there is some interpretation/analysis of this).

---

### Official Review · Reviewer_BzzB · 2025-10-29

**Soundness:** 2
**Presentation:** 1
**Contribution:** 2
**Rating:** 0
**Confidence:** 3

**Summary:**

The paper explores whether training data entropy or compressibility can serve as a proxy for LLM memorization difficulty. The authors propose the Entropy–Memorization (EM) Law, stating that the entropy of training data (computed via token-level statistics) correlates linearly with a memorization score (computed using Levenshtein distance). They perform experiments on multiple open LLMs (OLMo, OpenLLaMA, Pythia) and datasets, finding a strong linear correlation (Pearson’s correlation $\sim$ 0.9).

**Strengths:**

1. The paper explored a different perspective of connecting intrinsic property of training data to memorization.
2. Experiments have been conducted on multiple LLMs.

**Weaknesses:**

1. The paper needs an extensive amount of re-writing. It reads more like an exploratory report at present. There is lack of coherence across all the sections and the writing of the paper feels like has been done in a rush. For example,

- Page 2 line 107 - "Choices of LLM and Training Corpus lama (Geng & Liu, 2023)," -- this ends here, followed by the next line on the next page as "We selected four family of pre-trained only LLMs:.."

- Line 122 - "...we repeatedly randomly sample a sequence (length > |p+ s|) from the dataset until the number reaches the required number." -- What is meant by number reaching the required number?

2. Line 77 states that "It suggests that higher entropy correlates with a higher memorization score in LLMs." -- does the setup really hint at that? Doesn't the increase in distance indicate that as entropy increases, the model reproduces the data less accurately? So higher-entropy data are harder to memorize, and the model tends to fail on them, which is intuitive as well. It seems like more than higher memorization score, this metric hints at how difficult the memorization is. The paper began with memorization difficulty and gradually entered into memorization score for some reason, which is not clear.

3. The connection between data entropy/compressibility and memorization has been suggested in [a]. It is not clear how different things in this paper are with the cited paper.

4. There are no other quantitative metrics used like perplexity for comparison with entropy.

5. The pseudcode of algorithms are provided with no explanation provided. The experiment section is written in a form of a list -- one after another finding, with lack of intuition or reasoning as to why one would be interested in it.



[a] Carlini, Nicholas, et al. "Extracting training data from large language models." 30th USENIX security symposium (USENIX Security 21). 2021.

**Questions:**

Same as Weaknesses above and the following:

1. Since edit distance doesn't capture semantic similarity/dissimilarity, how will this method perform if the Levenshtein distance is changed to some other metrics like BertScore or some other form of embedding?

2. What theoretical explanation justifies calling it an Entropy-Memorization law rather than an empirical correlation?

---

### Note · Authors · 2025-11-18

I have read and agree with the venue's withdrawal policy on behalf of myself and my co-authors.